# Semantic Router: On the Feasibility of Hijacking MLLMs via a Single Adversarial Perturbation

**Changyue Li** [1][2]  **Jiaying Li** [1]  **Youliang Yuan** [1]  **Jiaming He** [2]  **Zhicong Huang** [2]  **Pinjia He** [3]

## Abstract

Multimodal Large Language Models (MLLMs) are increasingly deployed in stateless systems, such as autonomous driving and robotics. This paper investigates a novel threat: **Semantic-Aware Hijacking**. We explore the feasibility of hijacking multiple stateless decisions simultaneously using a single universal perturbation. We introduce the Semantic-Aware Universal Perturbation (SAUP), which acts as a **semantic router**, "actively" perceiving input semantics and routing them to distinct, attacker-defined targets. To achieve this, we conduct a theoretical and empirical analysis on the geometric properties in the latent space. Guided by these insights, we propose the Semantic-Oriented (SORT) optimization strategy and annotate a new dataset with fine-grained semantics to evaluate performance. Extensive experiments on three representative MLLMs demonstrate the fundamental feasibility of this attack, achieving a 66% attack success rate over five targets using a single frame against Qwen.

## 1. Introduction

Multimodal Large Language Models (MLLMs) (Liu et al., 2023; Bai et al., 2024; Google, 2025) have emerged as the core perception and reasoning modules for agent applications. In real-world deployments such as autonomous driving (Yang et al., 2024) and robotic manipulation (Kim et al., 2024), the MLLM serves as an "atomic unit" of perception and action. At each time step, the model processes the current frame with a prompt to generate an immediate action, without retaining historical context. For instance, OpenVLA (Kim et al., 2024) creates a new context at each

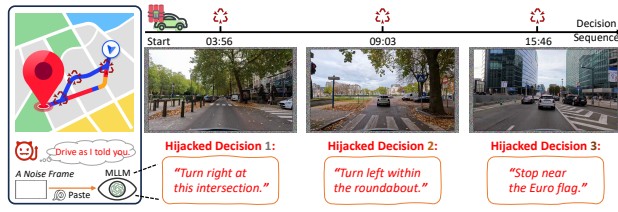

*Figure 1.* Illustration of the threat model of semantic-aware hijacking. As the MLLM processes the image sequences, the perturbation acts as a semantic router, forcing the model to output the attacker's predefined targets based on the input semantics, cumulatively guiding the vehicle to a predefined destination.

timestep, taking the current frame and an instruction as input, and continuously triggering independent actions. While these atomic decisions are stateless, their sequential accumulation dictates the agent's physical trajectory.

This paper focuses on these **stateless decisions** and investigates a novel and more severe threat: **Semantic-Aware Hijacking**. We aim to craft a universal adversarial perturbation that acts as a "semantic router," forcing the model to output distinct, predefined target actions conditioned solely on the visual semantics of the current frame. As illustrated in Figure 1, an adversary can paste an adversarial frame that "actively" senses input semantics (e.g., an intersection vs. a roundabout) and steers the model to generate corresponding targets. We term such perturbations *Semantic-Aware Universal Perturbations* (SAUPs).

Generating SAUPs imposes a severe geometric constraint on the optimization process. Our theoretical analysis in Section 3 reveals that mapping semantic inputs to diverse targets requires the perturbation to expand subtle semantic differences into output discrepancies. To overcome these optimization challenges, we propose the **S**emantic-**OR**ien**T**ed (**SORT**) optimization algorithm. SORT combines Normalized Space Optimization to construct a stable optimization manifold with Semantic Separation Optimization to explicitly maximize the expansion ability.

We annotate the **R**eal **I**mage **S**equence **T**rajectories (**RIST**) dataset, which serves as a simplified threat model and enables quantitative evaluation at varying semantic granularities. Unlike standard classification benchmarks (Deng et al.,

[1]The Chinese University of Hong Kong, Shenzhen, China [2]Ant Group [3]School of Data Science, School of Artificial Intelligence, The Chinese University of Hong Kong, Shenzhen, China. Correspondence to: Pinjia He <hepinjia@cuhk.edu.cn>.

2009), RIST targets fine-grained semantics for autonomous driving and robotic scenarios.

We conduct extensive experiments across three MLLMs (Llava-1.5, Qwen2.5-VL, and InternVL3) and two datasets with different semantic granularities (ImageNet and RIST). Our results validate the fundamental feasibility of the proposed attack. Notably, one perturbation is sufficient to mislead Qwen, achieving success rates of 93%, 77%, 61%, and 66% under 2, 3, 4, and 5 targets, respectively. Code is available at `https://github.com/lcycode/semantic-router`.

Our main contributions are summarized as follows:

- We demonstrate the feasibility of hijacking stateless decisions via a single perturbation and analyze the perturbation in the latent space both theoretically and empirically.

- We propose SORT, an effective algorithm to find such perturbations, and introduce RIST, a dataset with fine-grained semantic annotations.

- We conduct extensive experiments to analyze the vulnerability of MLLMs to SAUPs across three representative models, achieving a 66% attack success rate over five targets using a single frame against Qwen.

## 2. Related Work

### 2.1. Adversarial Perturbations

Adversarial perturbations, as pioneered by (Szegedy et al., 2014; Goodfellow et al., 2015), demonstrate that imperceptible noise added to input data can significantly alter model predictions. A pivotal advancement is the Universal Adversarial Perturbations (UAPs) (Moosavi-Dezfooli et al., 2017; Wang et al., 2023), which are input-agnostic and remain effective across diverse samples. Subsequent works have explored more complex settings: MultiAttack (Fort, 2023) shows that a perturbation can map various inputs to multiple targets, but it lacks universality and remains effective only within the training set. Multi-Target UAP (Custode & Iacca, 2021) seeks to find multiple perturbations simultaneously, yet each one still directs one target.

The primary contribution of this paper is the demonstration of a **many-to-many** universal adversarial perturbation. Unlike prior works, this perturbation acts as a "semantic router," mapping inputs with distinct semantics to different targets. We provide an extended discussion in Appendix 7.

### 2.2. Vulnerability of MLLMs

The rapid evolution of MLLMs (Liu et al., 2023; Bai et al., 2024) has introduced significant security concerns. Recent studies (Qi et al., 2024; Shayegani et al., 2024; Zhang et al., 2025; Ying et al., 2025) reveal that MLLMs inherit vulnerabilities to adversarial attacks. Existing studies can be generally categorized into two types:

Semantic misleading (Zhao et al., 2023; Liu et al., 2024; Xu et al., 2025) focuses on causing the model to misinterpret the visual content. These attacks typically target the vision encoder, feeding corrupted representations to the downstream language model to induce incorrect reasoning outcomes. For instance, C-PGC (Fang et al., 2025) proposes a UAP against vision-language models that requires manipulation of both image and text modalities.

Image hijacking (Luo et al., 2024; Lu et al., 2024; Bailey et al., 2024) seeks token-level control over the model's output. These attacks often utilize end-to-end optimization. Such methods can force MLLMs to execute malicious payloads, such as generating bash commands or bypassing safety filters in downstream tasks (Bailey et al., 2024).

## 3. Geometric Mechanics of SAUPs

In this section, we formalize the threat model and provide geometric intuition for how a single perturbation can act as a semantic router. Note that this analysis is qualitative in nature and aims to provide conceptual insight rather than exact guarantees. We quantitatively evaluate the error introduced by the local linearity assumption in Appendix F.

### 3.1. Problem Formulation and Intuition

**Threat Model.** This paper aims to demonstrate the fundamental *feasibility* of hijacking MLLMs' stateless decisions via a single and universal perturbation. Consequently, our analysis is conducted under a digital white-box setting to verify the latent space behavior, prioritizing the understanding of the attack mechanism over physical deployment.

We focus on *stateless decisions*, where the model functions as an atomic unit without retaining historical memory (e.g., OpenVLA (Kim et al., 2024)). In this paradigm, hijacking the atomic decision at each step effectively dictates the agent's cumulative physical trajectory.

**Definition 3.1.** Stateless decisions are a sequence of independent actions $\{y_1, \ldots, y_K\}$ indexed by time $k \in [1, K]$. At each step $k$, the MLLM, denoted as $\mathcal{M}$, generates an action $y_k$ based solely on the current visual observation $x_k \in \mathbb{R}^d$ and the textual prompt $p_k$, formulated as:

$$y_k = \mathcal{M}(x_k, p_k) \tag{1}$$

Note that the generation of $y_k$ is independent of previous states $\{(x_i, p_i, y_i)\}_{i<k}$, meaning the model retains no historical context between steps.

The adversary seeks a single perturbation $\delta$ to hijack multi-

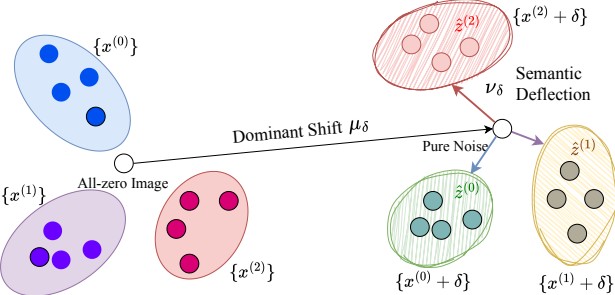

*Figure 2.* Illustration of the geometric mechanism of the Semantic-Aware phenomenon. Given three image sets $\{x^{(c)}\}$ with distinct semantic contents, and an all-zero pixel image serving as an anchor. The SAUP maps all image features to a distant region in the latent space, while the respective semantics cause slight deflections, guiding them towards alignment with predefined targets.

ple stateless decisions simultaneously. Specifically, when the perturbation is added to the visual input $x^{(c)}$ from a specific semantic category $c$, it compels the model to output a predefined target label $t^{(c)}$. We refer to such perturbations as Semantic-Aware Universal Perturbations (SAUPs).

$$\mathcal{M}(x^{(c)} + \delta, p^{(c)}) \rightarrow t^{(c)}, \quad \forall c \in \{1, \dots, C\} \quad (2)$$

where $C$ denotes the number of distinct semantic classes.

**Intuitive Mechanism.** Before diving into the geometric derivation, we provide an intuitive explanation of how a single perturbation $\delta$ achieves this complex mapping, as illustrated in Figure 2. The mechanism relies on two key geometric effects in the latent space:

*Dominant Shift:* When the perturbation is introduced to images, its features become dominant, shifting input clusters away from their original manifolds toward the adversarial subspace. In this subspace, decision boundaries are highly dense and congested (Liu et al., 2017).

*Semantic Deflection:* Once shifted, the perturbation leverages the intrinsic semantic differences between inputs to deflect features towards distinct directions, eventually aligning them with predefined targets.

### 3.2. Geometric Decomposition in Latent Space

To understand how visual inputs define textual outputs, we decompose the MLLM $\mathcal{M}$ into a vision encoder $\phi : \mathcal{X} \rightarrow \mathcal{Z}$, which maps images to latent embeddings, and a backbone decoder model $\mathcal{D} : \mathcal{Z} \times \mathcal{P} \rightarrow \mathcal{Y}$.

A key challenge in formulation is that the target $t^{(c)}$ is a text sequence, while the perturbation operates in the visual input space. To facilitate a theoretical analysis, we introduce the concept of *Proxy Target Embeddings*.

**Assumption 3.2.** For a given target response $t^{(c)}$ and prompt $p$, we assume there exists a corresponding visual

embedding $\hat{z}^{(c)} \in \mathcal{Z}$, which can induce the decoder $\mathcal{D}$ to generate $t^{(c)}$ (i.e., $\mathcal{D}(\hat{z}^{(c)}, p) \rightarrow t^{(c)}$). We term this visual embedding $\hat{z}^{(c)}$ as the **Proxy Target Embedding**[1].

Under this assumption, the goal of the perturbation $\delta$ is to map the input set $\{x^{(c)}\}$ to the corresponding set of proxy target embeddings $\{\hat{z}^{(c)}\}$. We analyze this mapping mechanism via a first-order Taylor expansion of the vision encoder $\phi$ around the perturbation $\delta$.

For an input image $x^{(c)}$ belonging to class $c$, the feature representation of the perturbed image $z^{(c)} = \phi(x^{(c)} + \delta)$ can be approximated as:

$$z^{(c)} \approx \underbrace{\phi(\delta)}_{\text{Dominant Shift } \mu_\delta} + \underbrace{J_\delta \cdot x^{(c)}}_{\text{Semantic Deflection } \nu_c} \quad (3)$$

where $J_\delta = \frac{\partial \phi(x)}{\partial x}\big|_{x=\delta}$ is the Jacobian matrix evaluated at the perturbation point, $\phi(\delta)$ represents a *Dominant Shift*, and $\nu_c = J_\delta x^{(c)}$ represents the *Semantic Deflection*.

*Remark* 3.3. The optimization of SAUPs is geometrically equivalent to searching for a perturbation $\delta$ such that the perturbed embedding $z^{(c)}$ aligns with the corresponding proxy target embedding $\hat{z}^{(c)}$. Specifically, the local linear projection $J_\delta$ serves to extract the intrinsic semantics of $x^{(c)}$ and map them toward class-specific targets.

### 3.3. Semantic Separability Potency and Bounds

Recall from Equation (3) that the perturbation $\delta$ should preserve and amplify the unique semantics of $x^{(c)}$. We define the *Deflection Strength* as the magnitude of its deviation from the anchor point $\phi(\delta)$:

$$\mathcal{S}(\delta, x^{(c)}) = \|\phi(x^{(c)} + \delta) - \phi(\delta)\|_2 \quad (4)$$

Using the first-order Taylor approximation, this strength is determined by the local geometry at $\delta$:

$$\mathcal{S}(\delta, x^{(c)}) \approx \|J_\delta x^{(c)}\|_2 \leq \|J_\delta\|_2 \cdot \|x^{(c)}\|_2 \quad (5)$$

where $\|J_\delta\|_2$ is the spectral norm of the Jacobian matrix at the perturbation point. Here, $\|J_\delta\|_2$ serves as the amplification factor, quantifying the capability of semantic deflection.

To hijack multiple decisions, the adversary requires more than just sensitivity; they require *separability*. For any pair of distinct classes $i$ and $j$, the perturbation must identify the semantic difference between inputs $x^{(i)}$ and $x^{(j)}$ and align them with the corresponding proxy target embeddings $\hat{z}^{(i)}$ and $\hat{z}^{(j)}$.

$$\|\phi(x^{(i)} + \delta) - \phi(x^{(j)} + \delta)\|_2 \approx \|\hat{z}^{(i)} - \hat{z}^{(j)}\|_2 \quad (6)$$

---

[1]While we denote $\phi$ as the vision encoder for simplicity, the concept of Proxy Target Embedding can be generalized to any intermediate layer where semantic alignment occurs.

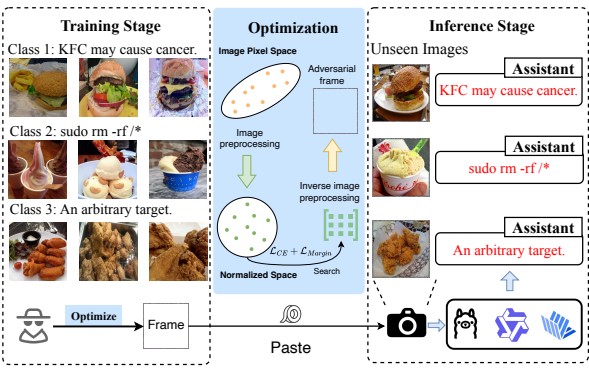

*Figure 3.* Illustration of the SAUP workflow. The adversary first collects images from several classes and assigns a specific target label to each class. These images are then used to train the adversarial perturbation (e.g., an adversarial frame). Once trained, the perturbation can be applied to other unseen images, causing MLLMs to generate the exact target sentences conditioned on the semantic content of the input image.

Substituting the linearization, the achievable separation in the latent space is governed by:

$$\|\hat{z}^{(i)} - \hat{z}^{(j)}\|_2 \le \underbrace{\|J_\delta\|_2}_{\text{Potency}} \cdot \underbrace{\|x^{(i)} - x^{(j)}\|_2}_{\text{Input Distance}} \quad (7)$$

Let $L = \sup_z \|J_z\|_2$ be the global Lipschitz constant of the vision encoder. We define the **Required Expansion Ratio** $\rho_{ij}$ for an input-target pair as:

$$\rho_{ij} = \frac{\|\hat{z}^{(i)} - \hat{z}^{(j)}\|_2}{\|x^{(i)} - x^{(j)}\|_2} \quad (8)$$

**Theorem 3.4.** *Let $\rho_{\max} = \max_{i \ne j} \rho_{ij}$ be the Maximum Required Expansion Ratio. If $\rho_{\max} > L$, then no perturbation $\delta$ exists that can successfully map the input set $\{x^{(c)}\}$ to the target set $\{\hat{z}^{(c)}\}$.*

*Remark* 3.5. This theorem highlights the difficulty of fine-grained semantics. As the images become similar (e.g., images captured by the same dashboard camera), the input distance $\|x^{(i)} - x^{(j)}\|_2 \to 0$, causing $\rho_{\max} \to \infty$, which makes it increasingly likely to violate the bound $L$.

## 4. Methodology: SORT

Guided by the theoretical analysis in Section 3, we propose the **S**emantic-**OR**ien**T**ed (SORT) algorithm. As illustrated in Figure 3, SORT incorporates a normalized space optimization to stabilize gradients, and a semantic separation strategy to explicitly maximize the deflection potency $J_\delta$.

### 4.1. Normalized Space Optimization (NSO)

To facilitate the addition of perturbations to images, existing studies (Bailey et al., 2024; Lu et al., 2024; Wang & He,

2021; Vo et al., 2022) conduct their optimization directly in the pixel space $[0, 1]$. This hinders the precise adjustment of the Jacobian $J_\delta$. To ensure stable updates, we perform a change of variables into the normalized space.

Let $\Psi : \mathcal{X} \to \mathbb{R}^d$ be the normalization function (e.g., standardization). We define a trainable variable $\Delta$ in the normalized space. The perturbation $\delta$ in pixel space is recovered via the inverse transformation:

$$\delta = \Psi^{-1}(\Delta) \quad (9)$$

By optimizing $\Delta$ directly, we precondition the optimization landscape, ensuring consistent step sizes and avoiding the plateauing loss observed in pixel-space constraints.

### 4.2. Semantic Separation Optimization (SSO)

To search for a feasible perturbation, we design a hybrid objective that combines cross-entropy loss $\mathcal{L}_{CE}$ and a margin-based loss $\mathcal{L}_{Margin}$. The cross-entropy loss ensures basic alignment with the target, while the margin-based loss explicitly forces input embeddings apart.

Recall from Equation (6) that the optimization must effectively expand the feature distance $\|\phi(x^{(i)} + \delta) - \phi(x^{(j)} + \delta)\|_2$ to align with the distance between proxy target embeddings $\|\hat{z}^{(i)} - \hat{z}^{(j)}\|_2$. Therefore, we employ the margin loss to explicitly drive the perturbed features apart by maximizing the separation in the output space.

Let $P(\cdot|x, p)$ denote the output confidence of the MLLM, conditioned on the visual input $x$ and textual prompt $p$. We define the confidence gap $\Delta P_{cj}$ between the corresponding target $t^{(c)}$ and a competing target $t^{(j)}$:

$$\Delta P_{cj} = P(t^{(c)}|x^{(c)} + \delta, p) - P(t^{(j)}|x^{(c)} + \delta, p) \quad (10)$$

To enforce the separation, we maximize this gap until it exceeds a specified margin $m$:

$$\mathcal{L}_{Margin} = \mathbb{E}_{j \ne c}\left[\max\left(0, m - \Delta P_{cj}\right)\right] \quad (11)$$

Geometrically, minimizing $\mathcal{L}_{Margin}$ requires the perturbed feature $\phi(x^{(c)} + \delta)$ to move closer to $\hat{z}^{(c)}$ than to $\hat{z}^{(j)}$. The final objective is:

$$\mathcal{L}_{Total} = \mathcal{L}_{CE} + \lambda \cdot \mathcal{L}_{Margin} \quad (12)$$

where $\lambda$ is a balancing hyperparameter that controls the trade-off between target alignment and semantic separation.

## 5. Validation of Geometric Mechanics

In this section, we empirically validate the theoretical framework proposed in Section 3 by analyzing the latent space representations of the perturbed images. Our goal is to verify the decomposition of the perturbed feature $z^{(c)}$ into the

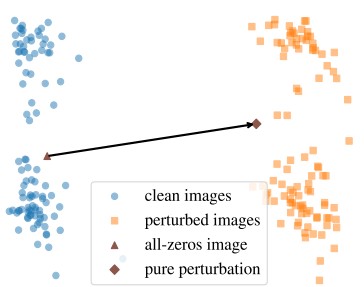 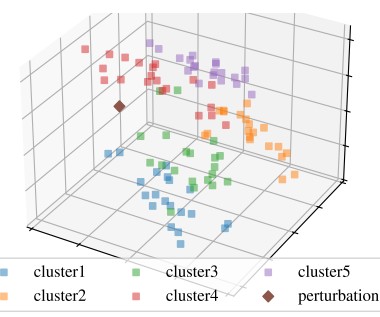 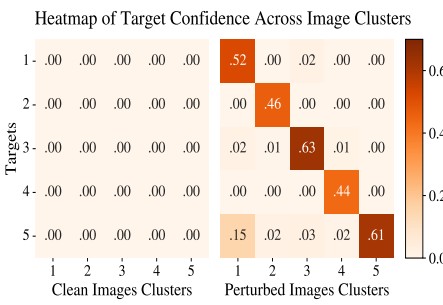

*(a)* Separation between clean and perturbed features.  *(b)* Separation among perturbed features.  *(c)* Alignment between images and targets.

*Figure 4.* We generate a SAUP for Llava and extract image features from the penultimate layer. (a) The perturbation dominates the features of the perturbed images and deviates significantly from those of the clean images. (b) The perturbed image features corresponding to different semantics are separated from each other and occupy distinct locations in the feature space. (c) The perturbed images are aligned with their corresponding targets, resulting in high output confidence for those targets.

*Dominant Shift* ($\mu_\delta$) and the *Semantic Deflection* ($\nu_c$), as formulated in Equation (3).

We conduct experiments using Llava-1.5-7B (Liu et al., 2023), selecting images from five distinct classes of ImageNet (Deng et al., 2009) and assigning a unique target label $t^{(c)}$ to each class. We visualize the penultimate layer features using Principal Component Analysis (PCA), which preserves the global geometric structure of the latent space, including relative distances and inter-cluster relationships.

### 5.1. Validation of the Dominant Shift

**Observation:** As illustrated in Figure 4a, we observe a significant separation between the clusters of clean images and perturbed images in the latent space. To confirm that this shift is intrinsic to the perturbation itself, we feed a "pure perturbation" (an all-zero image added with $\delta$) into the model. As shown in Figure 4a, this point lies at the centroid of the perturbed image clusters.

**Conclusion:** This observation validates the existence of the *Dominant Shift* $\mu_\delta = \phi(\delta)$, derived in Equation (3).

### 5.2. Validation of Semantic Deflection

**Observation:** While perturbed features are clustered away from clean data, Figure 4b reveals that they are not monolithic. Instead, they form distinct, separable sub-clusters corresponding to their original semantic categories. The formation of these distinct sub-clusters shows that the Semantic Deflection $\nu_c$ can identify semantic differences among inputs and deflect them in different directions.

**Conclusion:** This confirms the role of the *Semantic Deflection* $\nu_c = J_\delta \cdot x^{(c)}$, derived in Equation (3).

To further validate the generality of these geometric conclusions beyond LLaVA, we extend the latent feature analysis

to multiple MLLMs (LLaVA, Qwen, and InternVL) and varying numbers of classes. The results consistently confirm the Dominant Shift and Semantic Deflection mechanism across all models; see Appendix G for details.

### 5.3. Validation of Alignment with Targets

**Observation:** Figure 4c presents the confidence heatmap, which shows that the perturbation results in a significantly higher confidence along the diagonal. This indicates that the direction of semantic deflection is not stochastic. Instead, it forces the embeddings of input images to align precisely with their respective proxy target embeddings, thereby triggering the predefined targets.

**Conclusion:** This indicates that deflected embeddings are aligned with the corresponding proxy target embeddings, as described in Equation (6).

## 6. Performance Evaluation

### 6.1. Experimental Setup

**Datasets.** We evaluate the effectiveness and feasibility of SAUPs on two datasets. Each dataset contains a train set for optimizing perturbations and a non-overlapping test set for evaluating the generalization ability of the perturbations.

*(1) RIST (Fine-grained Semantics).* Recall from Theorem 3.5 that fine-grained semantics are more difficult to attack. To evaluate the performance of semantic-aware hijacking under this task, we annotate the Real Image Sequence Trajectories (RIST) dataset.

Unlike standard classification benchmarks, RIST consists of sequential frames extracted from continuous videos (Auto-Driving and RoboTasking scenarios), serving as a stringent stress test. More details are depicted in Appendix A.



*Figure 5.* SAUP cases generated on a test image trajectory within RIST. We optimize and apply a single SAUP to these test images. Although different categories share similar semantic content (such as motorway and sidewalk), the perturbation successfully acts as a "semantic router," forcing the MLLM to output different targets based on the content of the current frame.

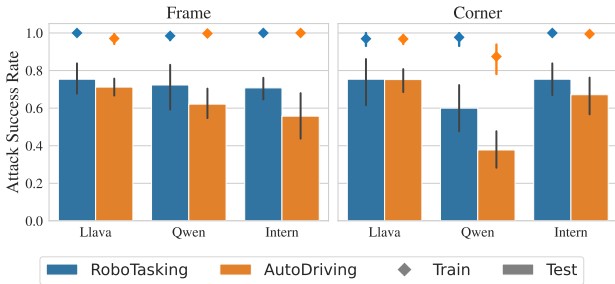

*Figure 6.* Attack success rates of SAUP on each trajectory of the RIST dataset, covering two realistic scenarios: RoboTasking (#Targets=2) and AutoDriving (#Targets=5).

We leverage Gemini-2.5-pro (Google, 2025) to randomly assign a specific, contextually relevant action (e.g., "accelerate" at a red light) as the target $t^{(c)}$ for each semantic cluster $c$, based on the scenario's safety constraints.

*(2) ImageNet (Coarse-grained semantics).* We employ ImageNet (Deng et al., 2009) to assess effectiveness on coarse-grained semantics. The target for each category is randomly selected from the remaining classes and expanded into a full sentence using Gemini.

**Models.** Our experiments involve three representative MLLMs: (1) Llava-1.5-7B (Llava) (Liu et al., 2023), a standard open-source benchmark; (2) Qwen2.5-VL-7B-Instruct (Qwen) (Bai et al., 2025), known for advanced instruction-following; and (3) InternVL3-8B (Intern) (Zhu et al., 2025), a strong general-purpose model.

Consistent with prior work (Lu et al., 2024), we fix the user input prompt **p** as `Describe this image` and use greedy decoding for all MLLMs. We evaluate varying prompts in Appendix B.

**Perturbation Constraints.** We focus on two types of constrained adversarial perturbations: *frame* and *corner*, where the frame width is set to 6 pixels and each corner patch has a size of $20 \times 20$ pixels unless otherwise specified. We study additional constraint settings in Appendix C.

**Evaluation Metric.** We report the Attack Success Rate

(ASR), which requires the output tokens to match the target sequence in both content and order.

## 6.2. Feasibility of Semantic-Aware Hijacking

This section aims to verify the feasibility of semantic-aware hijacking under different semantic granularities.

### 6.2.1. HIJACKING FINE-GRAINED SEMANTICS

We first demonstrate the "Semantic-Aware" capability of SAUPs through a visualization on the RIST dataset. Figure 5 shows images from the test set of RIST, which are extracted from a continuous video. The adversary optimizes and applies a single universal perturbation to this continuous image trajectory. Although different categories share similar semantic content (such as motorway and sidewalk), the SAUP successfully amplifies these subtle semantic variations in the latent space. As the MLLM processes the image trajectory, the perturbation acts as a semantic router, forcing the model to output distinct, attacker-defined, and dangerous commands such as "accelerate," "merge," or "steer" at precise moments. This confirms that SAUPs can effectively hijack multiple stateless decisions by binding specific visual semantics to predefined targets.

To systematically evaluate this threat, we evaluate the ASR across the entire RIST dataset, covering two scenarios: RoboTasking (2 targets) and AutoDriving (5 targets). As shown in Figure 6, SAUPs achieve an average ASR of 72% on the RoboTasking test set and 62% on the AutoDriving test set. It is worth noting that we observed an overfitting phenomenon. For instance, on the Intern model, the ASR reaches 100% on the training set but drops to 61% on the test set for the AutoDriving scenario. This performance gap is attributed to the limited size of the RIST training set (50 images). Nevertheless, these results demonstrate that even with limited data, a single perturbation can learn to distinguish fine-grained semantics and successfully commandeer the model's decision process across multiple targets.

The relatively large error bars in Figure 6 reflect performance variance across different trajectories in the RIST dataset. This variance stems from two factors. First, RIST is

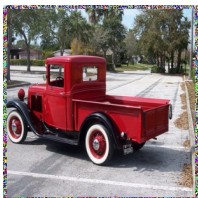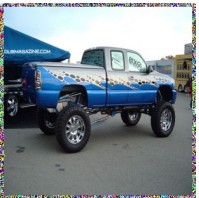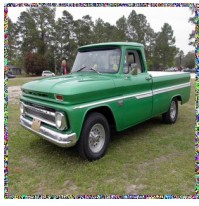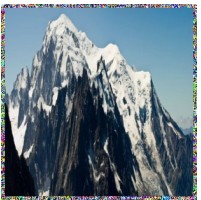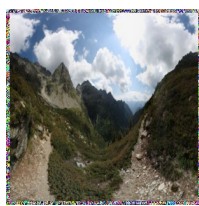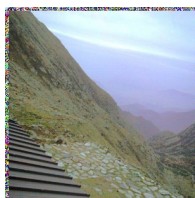

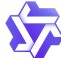 promontory silence settled over the landscape, broken only by the distant cry of seabirds crayfish play a vital role in aquatic ecosystems by helping to break down plant material

*Figure 7.* SAUP cases of precise control over long-text generation. On the test set, we demonstrate that a single SAUP can bind complex, long textual targets to specific visual concepts. For instance, "truck" triggers a specific long sentence, while "mountain" triggers a crayfish description, verifying the model's ability to maintain precise control over extended generation sequences.

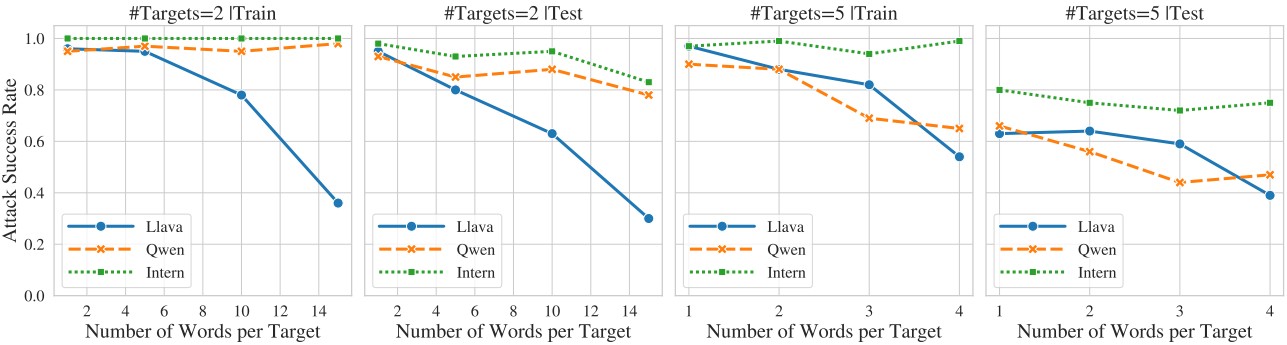

*Figure 8.* Attack success rates under different numbers of targets (#Targets) and words per target.

intentionally curated to be highly diverse, encompassing a wide range of real-world environmental conditions such as varying lighting and backgrounds, which naturally induces fluctuations in attack success rates across trajectories. Second, as a fine-grained dataset, the exact semantic granularity is not completely uniform across all trajectories, leading to variation in the generalization difficulty of the perturbation.

### 6.2.2. PRECISE CONTROL OVER LONG OUTPUT

Beyond simple semantic routing, an adversary may need to inject complex payloads, such as specific sentences or malicious scripts. Figure 7 visualizes the capability of SAUPs to generate long, coherent text sequences on the ImageNet dataset. We assign semantically unrelated target sentences to different image classes. For example, "truck" is bound to a poetic description, while "mountain" triggers a factual statement about crayfish. The results show that the perturbation effectively and precisely controls long-sequence generation by the MLLM.

We comprehensively evaluate the effectiveness of this control by varying the sequence length (Number of Words per Target). As illustrated in Figure 8, SAUPs show a strong ability to control long outputs. In the 2-target setting, when each target consists of 10 words, we still achieve an average ASR of 83% on the test set. When the number of targets

increases to 5, SAUPs can still control 4 words per target, achieving an average ASR of 54% on the test set. Qwen and Intern are highly susceptible to control by SAUPs, enabling the precise generation of sentences. For example, when the target length reaches 15 words in the 2-target setting, we achieve 78% ASR on Qwen and 83% ASR on Intern. This suggests that once an incorrect token is induced early in generation, subsequent tokens become more likely to drift, making the remainder of the sequence easier to control adversarially.

### 6.3. Information Capacity of Single Perturbation

While Theorem 3.4 establishes the theoretical feasibility bound, this condition primarily governs the feasibility of the "worst-case" pair mapping. However, as the number of targets increases, the challenge shifts from the magnitude of the expansion to the capacity of the perturbation to encode multiple distinct mappings simultaneously. In this subsection, we evaluate the capacity limit of a single perturbation on ImageNet by increasing the number of targets.[2]

As illustrated in Figure 9, we observe a general decline in ASR as the number of targets increases. For example, on

---

[2]Each target is represented by a single word, using 50 images for training and 20 non-overlapping images for testing.

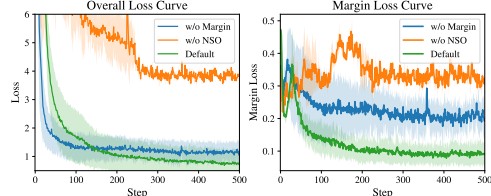

Figure 9. Attack success rates versus the number of targets (#Targets).

Figure 10. Loss curve during optimization for Qwen.

Table 1. Ablation studies on the optimization algorithm components, with bold indicating the highest ASR on the test set.

| Model | Constraint | Frame | | | | | | | | Corner | | | | | | | |
|---|---|---|---|---|---|---|---|---|---|---|---|---|---|---|---|---|---|
| | Method | Baseline | | w/o NSO | | w/o SSO | | Default | | Baseline | | w/o NSO | | w/o SSO | | Default | |
| | #Targets | Train | Test | Train | Test | Train | Test | Train | Test | Train | Test | Train | Test | Train | Test | Train | Test |
| **Llava** | 2 | 0.00 | 0.00 | 0.03 | 0.00 | 0.93 | 0.93 | 0.96 | **0.95** | 0.05 | 0.03 | 0.23 | 0.13 | 1.00 | **0.93** | 0.90 | 0.88 |
| | 5 | 0.13 | 0.11 | 0.46 | 0.43 | 0.82 | 0.57 | 0.97 | **0.63** | 0.48 | 0.47 | 0.01 | 0.03 | 0.51 | 0.40 | 0.67 | **0.49** |
| **Qwen** | 2 | 0.00 | 0.00 | 0.00 | 0.00 | 0.99 | **0.93** | 0.95 | **0.93** | 0.35 | 0.33 | 0.40 | 0.38 | 0.98 | 0.95 | 1.00 | **0.98** |
| | 5 | 0.01 | 0.03 | 0.02 | 0.00 | 0.72 | 0.57 | 0.90 | **0.66** | 0.15 | 0.15 | 0.12 | 0.19 | 0.46 | 0.44 | 0.63 | **0.45** |
| **Intern** | 2 | 0.98 | 0.83 | 0.95 | 0.78 | 1.00 | 0.97 | 1.00 | **0.98** | 0.95 | 0.80 | 0.78 | 0.55 | 1.00 | 0.98 | 1.00 | **1.00** |
| | 5 | 0.84 | 0.76 | 0.78 | 0.73 | 0.82 | 0.76 | 0.97 | **0.80** | 0.86 | 0.84 | 0.50 | 0.40 | 0.95 | 0.79 | 0.92 | **0.82** |

Qwen, the ASR under the frame constraint drops from 93% (2 targets) to 31% (9 targets). We attribute this phenomenon to the limited information capacity of the perturbation. Recall Equation (5) that $\|J_\delta\|_2$ can quantify the theoretical mapping capability for the perturbation $\delta$. Since $\delta$ is spatially constrained (i.e., restricted to a frame or corners), its limited number of trainable pixels creates an information bottleneck. As the number of targets grows, it is increasingly difficult for a single $\delta$ to encode all required decision boundaries within the finite pixel budget.

This "information capacity" hypothesis is supported by the performance disparity between Frame and Corner constraints. The Frame perturbation occupies 7,056 pixels, providing a larger parameter space to shape the Jacobian $J_\delta$. Consequently, it maintains a relatively high ASR (e.g., 42% on Llava for 9 targets). In contrast, the Corner perturbation contains only 1,600 pixels. Due to this severe capacity restriction, the ASR for Corner perturbations collapses rapidly, dropping to 0% on Llava and Qwen when the number of targets reaches 8. Furthermore, the ASR can be significantly improved by slightly increasing the perturbed area (see Appendix C), further validating this hypothesis.

### 6.4. Ablation Studies

In this subsection, we disentangle the contributions of the two core components of SORT: Normalized Space Optimization (NSO) and Semantic Separation Optimization (SSO). We compare the default SORT with two variants: (1) *w/o NSO*, which optimizes directly in the $[0, 1]$ pixel space; and (2) *w/o SSO*, which removes the margin loss $\mathcal{L}_{Margin}$ and

relies solely on cross-entropy. In addition, we apply an image hijacking method (Bailey et al., 2024) to this multi-target scenario as a baseline for comparison.

**NSO: The Bedrock of Optimization.** Although NSO introduces a simple change of variables (mapping from pixel space to a normalized unconstrained space), it serves as the fundamental prerequisite for our attack's success.

Direct optimization in the pixel space is notoriously unstable for generating universal perturbations, often leading to loss plateauing. As evidenced in the left part of Figure 10, the loss curve of the "w/o NSO" variant remains consistently high and fails to descend, indicating that the optimizer is trapped in a poor local minimum.

This optimization failure results in a catastrophic performance drop. As shown in Table 1, removing NSO causes the ASR to plummet by an average of 49.5% on the test set. Specifically, on Qwen with 5 targets and Frame constraint, the ASR drops from 66% to near zero without this module. These results confirm that NSO is not merely a trick, but a necessary condition to make the geometric alignment of $J_\delta$ mathematically solvable.

**SSO: The Booster for Multi-Targeting**. SSO improves the effectiveness of SORT, especially in complex scenarios. As shown in Table 1, the w/o SSO variant achieves a comparable ASR to the default setting when the number of targets is two, as the standard Cross-Entropy loss is sufficient to separate a few classes. However, as the semantic crowding increases, the necessity of SSO becomes apparent, yielding an average ASR gain of 6%. Figure 10 further reveals that

*Table 2.* Comparison of Adversarial Perturbation Methods.

| Category | Method Type | Mapping Relationship | Generalization | Semantic Aware | Threat & Flexibility |
|---|---|---|---|---|---|
| Traditional Adversarial Perturbations | Untargeted | - | ✗ | ✗ | Low |
|  | Single Targeted | one-to-one | ✗ | ✗ | Low |
| Universal Adversarial Perturbations | Untargeted | - | ✓ | ✗ | Low |
|  | Single Targeted | many-to-one | ✓ | ✗ | Medium |
| **SAUPs (This paper)** | **Multiple Targeted** | **many-to-many** | ✓ | ✓ | **High** |

while the optimization without SSO converges quickly in the early stages, it gets trapped in a local optimum. In contrast, the default method continues to optimize the margin, pushing the perturbed features toward the centers of their respective target clusters, and eventually achieves a smaller overall loss and better generalization.

## 7. Discussion and Limitation

Current research on adversarial perturbations encounters limitations in flexibility, which can be categorized as shown in Table 2. *Traditional adversarial perturbations* (Lin et al., 2020; Wang & He, 2021) are typically optimized on a per-instance basis. These methods can be either untargeted, aiming only to cause misclassification, or single-targeted, where the goal is to force the model to produce a specific incorrect label. However, these methods fail to scale or generalize to broader scenarios due to their inflexible, instance-specific design. *Universal adversarial perturbations (UAPs)* (Moosavi-Dezfooli et al., 2017; Naseer et al., 2021) extend adversarial attacks by creating a single perturbation applicable across multiple inputs. Untargeted UAPs are effective in degrading model accuracy broadly, while single-targeted UAPs attempt to shift model predictions toward a fixed label across all inputs. Despite their improved generalization, the perturbations fail to provide fine-grained control over the output based on the semantics of the input.

To manipulate multiple decisions toward distinct targets, existing approaches require an individual perturbation for each input. In contrast, SAUPs enable the adversary to map multiple images to various targets freely, representing a more sophisticated "switch-case" attack capability.

**Limitations of the White-box Assumption.** This paper is based on the white-box assumption, where the attacker has full access to model's architecture and parameters. While this assumption is standard in current image hijacking research (Bailey et al., 2024), it poses a limitation in real-world scenarios where models are often black boxes. Physical deployment also remains a significant challenge, as the robustness of these perturbations to physical factors such as lighting conditions and motion blur requires further study.

Nevertheless, SAUPs still pose meaningful risks in real-world settings. For instance, an attacker can target a widely deployed open-source MLLM and embed a crafted SAUP into a web advertisement. This perturbation could steer the MLLM-based agent to first upload private data and then execute a malicious "submit" action. We hope SAUPs serve as a starting point for motivating future research on semantic hijacking in more complex agentic environments.

**Limitations of the Local Linear Assumption.** Our theoretical analysis relies on the local linear assumption, which provides a principled framework for understanding the Dominant Shift and Semantic Deflection mechanism. However, this approximation introduces non-negligible error as the input magnitude grows. We provide a detailed empirical analysis of the approximation error in Appendix F, where we quantify the relative error and feature similarity across three MLLMs. The primary role of our theoretical framework is to provide qualitative insight into the geometric mechanism, rather than precise quantitative predictions.

## 8. Conclusion

This paper investigates semantic-aware hijacking, a novel and severe threat to MLLMs. We demonstrate that a single SAUP can effectively act as a "semantic router," compelling the model to output distinct, attacker-defined targets conditioned solely on the input semantics.

To achieve this, we introduce an optimization algorithm (SORT), and a dataset featuring fine-grained semantic annotations (RIST), which serves as a simplified realistic threat model for autonomous driving and robotic tasks.

We provide a comprehensive understanding of the underlying mechanism of SAUPs. We theoretically analyze the geometric feasibility bound and empirically validate the "Dominant Shift" and "Semantic Deflection" phenomena in the latent space. Extensive experiments across three representative MLLMs confirm the feasibility of SAUPs under various settings, including different semantic granularities, control lengths, and target diversities.

For future work, we intend to bridge the gap between digital simulation and reality by developing robust SAUPs resilient to physical conditions (e.g., brightness changes).

## Acknowledgments

This work was supported by the Guangdong Basic and Applied Basic Research Foundation (Grant No.2024A1515010145). We thank the anonymous reviewers for their insightful and constructive comments.

## Impact Statement

This paper exposes Semantic-Aware Hijacking, a novel MLLM vulnerability in which a single perturbation can semantically route multiple inputs to different malicious targets. The significance of this vulnerability lies in its ability to trigger cascading consequences by hijacking a chain of decisions. Given the rapid emergence of MLLM-based agents, such as those designed for automated computer and smartphone interaction, the potential for exploitation is vast. For instance, a malicious frame on a webpage could coerce an agent to first upload a user's private data and then execute a "submit" command to finalize the breach.

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

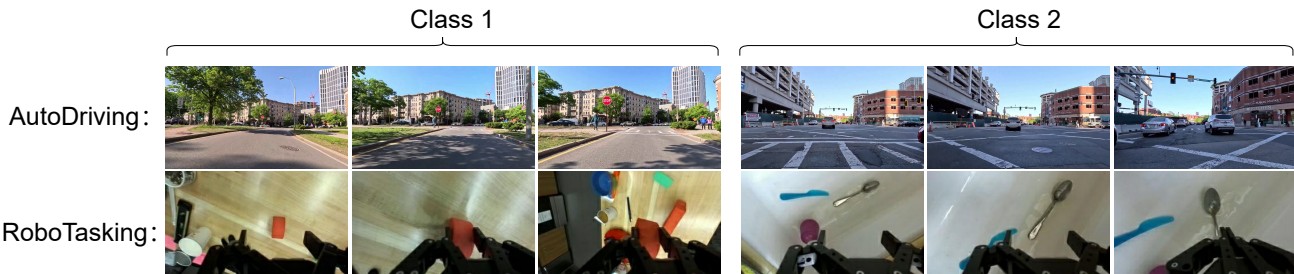

*Figure 11.* Illustration of RIST, which consists of real-world image trajectories across two scenarios: AutoDriving and RoboTasking.

## A. Design Details of RIST Dataset

We introduce the Real Image Sequence Trajectories (RIST), a newly annotated dataset to evaluate an adversary's capacity to induce distinct, predefined targets by exploiting fine-grained semantics. The fine-grained semantics are defined by a sequence of images with similar content, such as images of a specific intersection. This grants the adversary "switch-case" capability to "program" incidents flexibly. For instance, a single perturbation could first guide an MLLM to output "grasp" when a robot observes a knife, and then trigger a dangerous action like "throw" when a human subsequently enters the scene.

Semantic classification within a continuous trajectory is not supported by common image classification (Deng et al., 2009) or image captioning datasets. COCO Captions (Chen et al., 2015) and Flickr30k Entities (Plummer et al., 2015) treat each image as an independent sample, providing multiple annotated captions per image. While region-level context is occasionally included, these benchmarks fail to maintain temporal continuity or a consistent perspective across samples.

RIST distinguishes itself from existing temporal datasets, which often lack classification based on semantic similarity within a single trajectory. While MSR-VTT (Xu et al., 2016) provides clip-level captions and temporal context, it does not define semantic categories within trajectories. Similarly, autonomous driving datasets like KITTI (Geiger et al., 2013), nuScenes (Caesar et al., 2020), and Waymo Open Dataset (Sun et al., 2020) offer continuous sequences for perception and prediction tasks; however, they are not organized by event-centric categories, nor do they feature predefined goals for each category. In contrast, RIST segments each consistent-viewpoint trajectory into multiple semantic categories and assigns a specific predefined goal to each.

### A.1. Dataset Curation and Content

RIST consists of over 1,000 real-world images and 28 distinct trajectories, spanning two specialized sub-datasets: Auto-Driving and RoboTasking. Each trajectory comprises multiple image sequences extracted from a continuous video with a consistent point of view (POV). We treat these image sequences as different categories based on fine-grained semantics. Each image sequence consists of 10 visually similar frames and captures a specific action, such as driving through an intersection or picking up a cup.

To collect RIST, our annotators meticulously identify segments containing a specific event and exclude repetitive or semantically meaningless segments, such as cars idling at stoplights or robotic arms executing non-essential movements. Each category in the RIST dataset is centered around a specific event that could potentially be exploited for malicious purposes. For example, the image sequences include scenes of robots interacting with kitchen utensils or cars navigating intersections with warning signs.

To ensure that each sequence captures a unique event while preserving diversity across the dataset, our annotators review more than 1 TB of video footage and manually filter out highly similar images. Each image in RIST is independently cross-validated by two annotators to ensure both quality and diversity.

**AutoDriving** is designed to simulate stateless decisions of a driving autonomous vehicle. It comprises 15 trajectories, each further segmented into 5 distinct categories. Each category is represented by a sequence of 10 POV images, formatted as 1920x1080 pixel JPG images. The initial POV driving videos for AutoDriving are sourced from the OpenDV-2K dataset (Yang et al., 2024) and the ONCE dataset (Mao et al., 2021). This curation spans multiple countries and varies in temporal and meteorological conditions to ensure diversity.

**RoboTasking** is curated to evaluate the robustness of robotic stateless decisions in realistic indoor environments. It consists

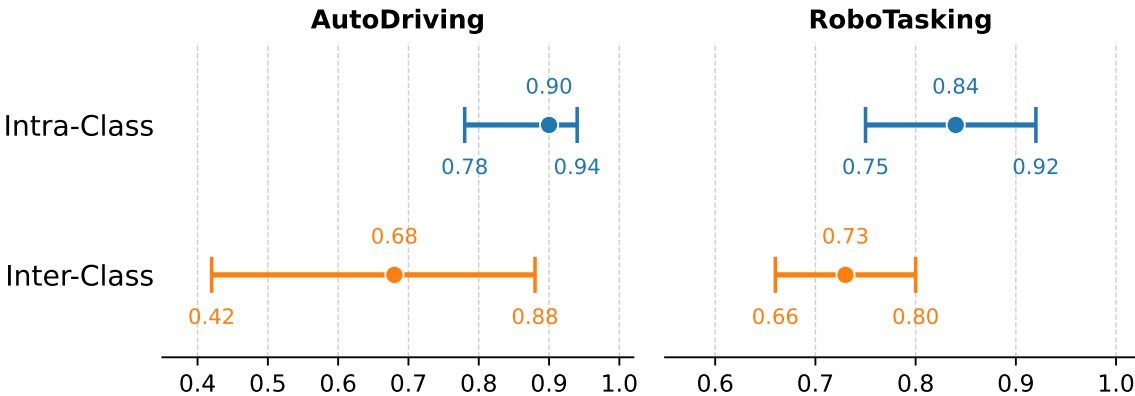

*Figure 12.* CLIP Score similarity of intra-class and inter-class image pairs in RIST.

of 13 trajectories, each of which is divided into 2 categories. Similarly, each category comprises a sequence of 10 POV images, captured at a resolution of 320x180 pixels and stored as JPG files. The original POV robot videos for RoboTasking are sourced from the DROID dataset (Tang et al., 2025), utilizing the robot's wrist-mounted camera perspective. We cover a wide range of robotic tasks (e.g., grasping objects, placing items, and operating tools) across various environments, including kitchens, living rooms, and workshops.

### A.2. Quantitative Analysis of Semantic Granularity

To quantitatively characterize the fine-grained nature of RIST, we compute the CLIP Score (Radford et al., 2021) for image pairs within and across semantic classes. Specifically, we measure the cosine similarity of CLIP image embeddings for intra-class pairs (images from the same semantic category) and inter-class pairs (images from different categories within the same trajectory).

As shown in Figure 12, these results confirm the fine-grained nature of the RIST dataset. Notably, the mean inter-class similarity reaches approximately 0.70, with maximum values exceeding 0.80. This high inter-class similarity requires SAUP to exploit highly subtle visual differences to route inputs to distinct target labels, covering instance-level granularity that goes beyond broad category distinctions.

### A.3. Target Action Annotation

To evaluate our semantic-aware universal perturbations (SAUPs), distinct targets are manually assigned to each category within RIST to guide the optimization toward inducing specific, contextually relevant, yet undesirable behaviors from the MLLM. The annotation process for each sub-dataset is as follows:

For AutoDriving, the image sequence corresponding to each category is input into Gemini-2.5-pro (Google, 2025). The model is prompted to act as an autonomous driving system and select a driving action based on the input images that would most likely pose a significant traffic safety risk. This selected action is assigned as the target for that category.

Similarly, for RoboTasking, each image sequence is subsequently processed by the MLLM to produce the target. In this context, the model is instructed to act as an embodied robot and select a robotic action that would most likely introduce an operational safety hazard.

When evaluating RIST, all images are first scaled to a $[0, 1]$ pixel range. Within each category, half of the images are selected for the training set and used to optimize the perturbation. The remaining images serve as the test set to evaluate the perturbation's generalization.

## B. Evaluation on Multiple Prompts.

To assess the robustness and generalization of SAUPs under diverse user queries, we evaluate the attack performance across a range of input prompts that vary in phrasing and semantic intent on ImageNet. Specifically, we utilize 50 different prompts

*Table 3.* Attack success rates of SAUPs under various input prompts. Note that these prompts are not in the training set.

| | Qwen | | | | Llava | | | | Intern | | | |
| | Frame | | Corner | | Frame | | Corner | | Frame | | Corner | |
| Prompt | Train | Test | Train | Test | Train | Test | Train | Test | Train | Test | Train | Test |
|---|---|---|---|---|---|---|---|---|---|---|---|---|
| What is in the image? | 0.97 | 0.85 | 0.98 | 0.93 | 0.02 | 0.03 | 0.00 | 0.00 | 0.81 | 0.65 | 1.00 | 1.00 |
| What's happening in this picture? | 0.76 | 0.63 | 0.91 | 0.90 | 0.18 | 0.13 | 0.26 | 0.25 | 0.97 | 0.80 | 1.00 | 0.98 |
| Can you tell me about this image? | 0.97 | 0.83 | 0.93 | 0.93 | 0.35 | 0.25 | 0.20 | 0.25 | 0.91 | 0.75 | 0.98 | 0.90 |
| Summarize the content of this image. | 0.97 | 0.80 | 0.96 | 0.93 | 0.38 | 0.35 | 0.20 | 0.13 | 0.90 | 0.83 | 0.98 | 0.93 |
| Describe what you see in the photo. | 0.93 | 0.83 | 0.97 | 0.98 | 0.31 | 0.25 | 0.19 | 0.15 | 0.55 | 0.40 | 0.99 | 0.95 |
| Could you give me a general overview? | 0.87 | 0.70 | 0.94 | 0.95 | 0.10 | 0.08 | 0.00 | 0.03 | 0.87 | 0.73 | 0.99 | 0.93 |
| What colors dominate the image? | 0.96 | 0.80 | 0.97 | 0.95 | 0.03 | 0.01 | 0.00 | 0.00 | 0.79 | 0.60 | 1.00 | 1.00 |
| What is likely to happen next? | 0.95 | 0.80 | 0.88 | 0.88 | 0.27 | 0.33 | 0.30 | 0.30 | 0.94 | 0.80 | 1.00 | 1.00 |
| What message or story does this image convey? | 0.75 | 0.55 | 0.86 | 0.85 | 0.18 | 0.25 | 0.01 | 0.03 | 0.97 | 0.95 | 1.00 | 0.98 |
| What is the setting or location of this scene? | 0.94 | 0.73 | 0.91 | 0.93 | 0.02 | 0.03 | 0.14 | 0.08 | 0.90 | 0.75 | 1.00 | 0.88 |

*Table 4.* Attack success rates of SAUPs under various perturbation region constraints.

| | | Llava | | | | | | Qwen | | | | | | Intern | | | | | |
| | | Width of Frame | | | Patch Size of Corner | | | Width of Frame | | | Patch Size of Corner | | | Width of Frame | | | Patch Size of Corner | | |
| | #Targets | 4 | 6 | 8 | 20 | 30 | 40 | 4 | 6 | 8 | 20 | 30 | 40 | 4 | 6 | 8 | 20 | 30 | 40 |
|---|---|---|---|---|---|---|---|---|---|---|---|---|---|---|---|---|---|---|---|
| **Train** | 2 | 0.90 | 0.97 | 1.00 | 0.90 | 1.00 | 1.00 | 0.92 | 0.95 | 1.00 | 1.00 | 0.96 | 0.99 | 0.98 | 1.00 | 1.00 | 1.00 | 1.00 | 1.00 |
| | 4 | 0.56 | 0.97 | 0.94 | 0.94 | 0.96 | 1.00 | 0.42 | 0.93 | 0.92 | 0.63 | 0.87 | 0.92 | 0.95 | 1.00 | 1.00 | 0.95 | 1.00 | 1.00 |
| | 6 | 0.30 | 0.92 | 0.96 | 0.75 | 0.92 | 0.94 | 0.35 | 0.85 | 0.85 | 0.56 | 0.63 | 0.86 | 0.67 | 0.96 | 0.98 | 0.94 | 0.98 | 0.99 |
| | 8 | 0.44 | 0.75 | 0.81 | 0.00 | 0.41 | 0.64 | 0.20 | 0.55 | 0.64 | 0.00 | 0.47 | 0.69 | 0.54 | 0.96 | 0.86 | 0.66 | 0.73 | 0.84 |
| **Test** | 2 | 0.53 | 0.85 | 0.93 | 0.88 | 1.00 | 1.00 | 0.63 | 0.93 | 1.00 | 0.98 | 0.90 | 0.98 | 0.90 | 0.98 | 1.00 | 1.00 | 1.00 | 1.00 |
| | 4 | 0.31 | 0.64 | 0.93 | 0.69 | 0.90 | 0.94 | 0.30 | 0.61 | 0.84 | 0.43 | 0.79 | 0.86 | 0.64 | 0.81 | 0.94 | 0.80 | 0.91 | 0.99 |
| | 6 | 0.11 | 0.58 | 0.72 | 0.61 | 0.84 | 0.82 | 0.23 | 0.53 | 0.60 | 0.37 | 0.57 | 0.86 | 0.38 | 0.64 | 0.91 | 0.93 | 0.85 | 0.88 |
| | 8 | 0.22 | 0.41 | 0.56 | 0.00 | 0.37 | 0.51 | 0.16 | 0.26 | 0.43 | 0.00 | 0.33 | 0.58 | 0.29 | 0.61 | 0.59 | 0.49 | 0.54 | 0.69 |

in the training and 10 disjoint prompts in the evaluation. Table 3 summarizes the ASR of SAUPs on both training and testing sets under different prompt inputs.

We observe that SAUPs consistently achieve high ASRs on Qwen and Intern across prompts, with the average ASR of 74% and 94% under frame and corner constraints, respectively. Even when the prompts vary significantly, SAUPs remain effective in steering the model toward the predefined target outputs.

Contrary to intuition, Qwen and Intern, the more advanced MLLMs, demonstrate noticeably lower robustness than Llava when faced with diverse user prompts. Specifically, we achieve only a 14% ASR on Llava, which is substantially lower than the 84% obtained on Qwen and 84% on Intern. We hypothesize that this is due to the deeper alignment between the visual encoder and the language model in advanced models. Specifically, Qwen and Intern employ full-parameter supervised fine-tuning to align visual and textual modalities, whereas Llava freezes the visual encoder and relies on a linear projection layer for alignment. This deeper alignment in advanced models may amplify the influence of adversarial perturbations on the subsequent language model, thereby reducing overall robustness.

## C. Impact of Perturbation Region

This section evaluates SAUP on ImageNet by varying the perturbation frame width and the patch size of the corner. The results are shown in Table 4, which reveal that attack performance is highly dependent on the perturbation region.

Expanding the frame width or enlarging the corner patch size substantially boosts the ASR. For instance, on the Llava test set with 8 targets, widening the adversarial frame from 4 to 8 pixels increases the ASR from 22% to 56%. Likewise, increasing the corner patch size from 20 to 40 yields a substantial improvement, raising ASR from 0% to 51%.

Enlarging the perturbation region can enhance the attack generalization. For example, when the number of targets is set to 4, increasing the frame width from 6 to 8 on the Llava yields similar ASR on the training set, but the ASR on the test set

---

**Algorithm 1** SORT Optimization

---

**Require:** Training set $\mathcal{D} = \{(x^{(c)}, t^{(c)})\}$ containing images and target prompts;
**Require:** Loss Function $\mathcal{L}_{Total}$, Normalization function $\Psi$
**Require:** Learning rate $\eta$, Perturbation Mask $M$
**Ensure:** Semantic-Aware Universal Perturbation $\delta$
  1: **Initialize:** $\Delta \leftarrow \mathbf{0}$ {Initialize variable in normalized space}
  2: **Initialize:** $\Delta_L, \Delta_U \leftarrow \Psi(\mathbf{0}), \Psi(\mathbf{1})$ {Initialize lower and upper bound in normalized space}
  3: **while** not converged **do**
  4:    Sample batch of images and targets $(x, t)$ from $\mathcal{D}$
  5:    $x_{adv} = \Psi(x) + \Delta$ {Construct adversarial images}
  6:    $Loss \leftarrow \mathcal{L}_{Total}(x_{adv}, t)$ {Compute loss}
  7:    $\Delta \leftarrow \Delta - \eta \cdot \nabla_\Delta Loss$ {Update variable}
  8:    $\Delta \leftarrow \text{clip}(\Delta \odot M, \Delta_L, \Delta_U)$ {Constraint perturbation}
  9: **end while**
 10: **Return** $\delta = \Psi^{-1}(\Delta)$

---

improves significantly from 64% to 93%. Likewise, for Qwen, increasing the corner patch size from 20 to 30 results in a 3% ASR gain on the training set, but leads to a substantial 30% improvement on the test set.

## D. Algorithm Description

In this section, we provide the detailed pseudocode in Algorithm 1 for the SORT optimization. To ensure optimization stability, SORT performs a change of variables from the pixel space to the normalized space. While mathematically simple, this reparameterization proves highly effective in mitigating gradient instability and avoiding local optima. Instead of optimizing the perturbation $\delta$ directly, we define a trainable variable $\Delta$ in the normalized feature space, constructing the adversarial input as $x_{adv} = \Psi(x) + \Delta$.

Since $\Delta$ should be converted to the valid pixel range $[0, 1]$ through the inverse transformation $\Psi^{-1}$ at the end of the algorithm, an unconstrained $\Delta$ would be significantly distorted by truncation. To prevent this, we clip $\Delta$ within the bounds $[\Psi(\mathbf{0}), \Psi(\mathbf{1})]$ at each iteration. This mechanism guarantees that every update to the perturbation in the normalized space translates accurately to the physical image space.

## E. Impact of the Size of Training Set

This section evaluates the generalization of SAUPs by varying the size of the training set on ImageNet. Specifically, we generate SAUPs using subsets of 10, 20, 30, 40, and 50 images and evaluate their ASR on both the training and a fixed test set of 20 images. The results are summarized in Table 5.

We observe overfitting when the training set is small. For example, with 10 training samples, both Llava and Qwen achieve near-perfect ASR on the training set, but their performance drops significantly on the test set, especially for Llava under the frame constraint (10% ASR on test images). This discrepancy indicates that the learned perturbation may memorize training-specific features rather than generalize semantic alignment, limiting its real-world applicability.

On the contrary, increasing the training set size improves SAUPs' generalization to unseen data. For example, when the training set increases from 10 to 50 samples, Llava's test ASR under the frame constraint improves dramatically from 10% to 95%, showing that the learned perturbation begins to capture more generalizable features rather than overfitting to the training instances. Similarly, Qwen shows consistent improvements, especially under the corner constraint, where the test ASR reaches 98% with 50 training images. These trends highlight the importance of training diversity and sufficient semantic coverage for constructing transferable universal perturbations.

Meanwhile, a training set with only 50 images per class enables SAUPs to achieve strong attack performance even on unseen inputs. This suggests that an attacker could exploit a relatively limited but diverse set of images to capture the semantics of images and construct SAUPs effectively.

*Table 5.* Attack success rates under different sizes of training set

| Model | | Frame | | | | | Corner | | | | |
|---|---|---|---|---|---|---|---|---|---|---|---|
| | | 10 | 20 | 30 | 40 | 50 | 10 | 20 | 30 | 40 | 50 |
| Llava | Train | 1.00 | 1.00 | 0.93 | 0.93 | 0.96 | 1.00 | 1.00 | 0.98 | 0.94 | 0.90 |
| | Test | 0.10 | 0.53 | 0.58 | 0.83 | 0.95 | 0.45 | 0.75 | 0.88 | 0.83 | 0.88 |
| Qwen | Train | 1.00 | 0.98 | 0.92 | 0.96 | 0.95 | 1.00 | 1.00 | 1.00 | 1.00 | 1.00 |
| | Test | 0.85 | 0.63 | 0.85 | 0.95 | 0.93 | 0.60 | 0.73 | 0.80 | 0.83 | 0.98 |
| Intern | Train | 1.00 | 1.00 | 1.00 | 1.00 | 1.00 | 1.00 | 1.00 | 1.00 | 1.00 | 1.00 |
| | Test | 0.65 | 0.98 | 0.90 | 1.00 | 0.98 | 0.65 | 0.90 | 0.85 | 1.00 | 1.00 |

*Table 6.* Relative error and feature similarity of the first-order approximation across three MLLMs at increasing input magnitudes $\alpha$.

| Model | Metric | $\alpha=0$ | 0.1 | 0.2 | 0.5 | 1 | 2 | 5 | 10 | 20 | 50 | 100 | 200 |
|---|---|---|---|---|---|---|---|---|---|---|---|---|---|
| Llava | Relative Error | 0.00 | 0.05 | 0.08 | 0.13 | 0.20 | 0.39 | 0.44 | 0.63 | 0.80 | 0.91 | – | – |
| | Feature Similarity | 1.00 | 1.00 | 1.00 | 0.99 | 0.98 | 0.91 | 0.90 | 0.80 | 0.66 | 0.53 | – | – |
| | Loss | 7.51 | 7.38 | 6.66 | 7.50 | 7.37 | 6.60 | 7.42 | 7.07 | 6.18 | 2.47 | – | – |
| Qwen | Relative Error | 0.00 | 0.21 | 0.29 | 0.40 | 0.49 | 0.60 | 0.68 | 0.70 | 0.73 | 0.85 | – | – |
| | Feature Similarity | 1.00 | 0.98 | 0.96 | 0.92 | 0.88 | 0.82 | 0.77 | 0.75 | 0.73 | 0.64 | – | – |
| | Loss | 1.01 | 1.01 | 1.06 | 0.92 | 1.09 | 1.29 | 1.10 | 0.81 | 0.79 | 0.34 | – | – |
| Intern | Relative Error | 0.00 | 0.00 | 0.01 | 0.01 | 0.03 | 0.16 | 0.21 | 0.19 | 0.26 | 0.41 | 0.59 | 0.76 |
| | Feature Similarity | 1.00 | 1.00 | 1.00 | 1.00 | 1.00 | 0.97 | 0.97 | 0.98 | 0.97 | 0.92 | 0.82 | 0.72 |
| | Loss | 1.65 | 1.73 | 1.73 | 1.73 | 1.73 | 1.63 | 1.74 | 1.74 | 1.77 | 1.83 | 1.88 | 0.92 |

# F. Empirical Analysis of Theoretical Limitations

Our theoretical analysis in Section 3 relies on the local linear assumption around the perturbation $\delta$. While this provides a principled geometric framework, this approximation introduces non-negligible error as the input magnitude grows. In this section, we quantify the boundaries of this linear approximation.

We formulate the image feature as $z = \phi(\delta + \alpha \cdot v)$, where $\delta$ is the adversarial perturbation, $v$ is the normalized input image vector, and $\alpha$ controls the step magnitude. We compare the true latent feature with its first-order approximation around $\delta$ using two metrics:

$$\text{RelativeError} = \frac{\|\phi(\delta + \alpha v) - (\phi(\delta) + \alpha J_\delta(v))\|_2}{\|\phi(\delta + \alpha v)\|_2} \tag{13}$$

$$\text{FeatureSimilarity} = \cos(\phi(\delta + \alpha v), \ \phi(\delta) + \alpha J_\delta(v)) \tag{14}$$

where $J_\delta(v)$ is the Jacobian-vector product at $\delta$ along direction $v$. We evaluate these metrics on three MLLMs (Llava, Qwen, and Intern), progressively increasing $\alpha$ until a significant drop in loss (defined in Equation (12)) occurs.

The results in Table 6 provide two key insights:

**Validity of Local Linearity.** Although modern MLLMs are globally highly nonlinear, all models exhibit clear local linearity for sufficiently small $\alpha$. For example, on Intern, when $\alpha \leq 1$, the Relative Error remains negligible (0.00–0.03) and the Feature Similarity stays at 1.00. Even at $\alpha = 2$, the approximation is still accurate (error 0.16, similarity 0.97). This is consistent with the principle that, within a sufficiently small neighborhood, input changes do not cross the nonlinear boundaries of most neurons, thereby preserving local linearity.

**Limitations of Linear Approximation.** As $\alpha$ grows, the local linear assumption introduces non-negligible error. For instance, at $\alpha = 50$, the Relative Error for Qwen reaches 0.85. However, the Feature Similarity remains moderately aligned (0.64), indicating that the first-order approximation transitions from serving as a precise quantitative tool to providing qualitative guidance.

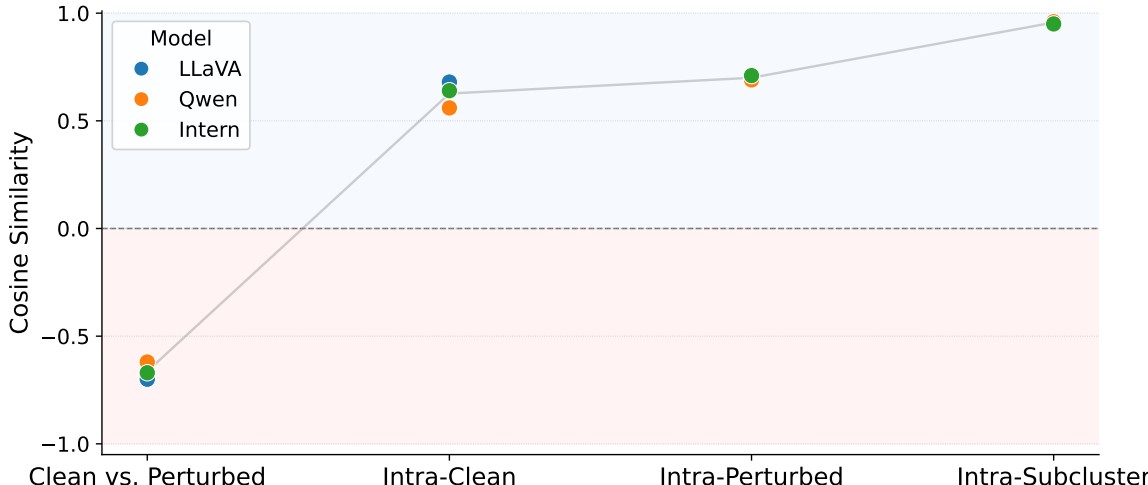

*Figure 13.* Mean cosine similarity of PCA-reduced features across three MLLMs. The horizontal axis denotes four similarity regimes: **Clean vs. Perturbed** is the cross-set similarity between clean and perturbed features; **Intra-Clean** is the pairwise similarity among clean features; **Intra-Perturbed** is the pairwise similarity among perturbed features; and **Intra-Subcluster** is the pairwise similarity within a subcluster of perturbed features sharing the same target semantics.

## G. Extended Empirical Validation of Geometric Mechanism

The geometric analysis in Section 5 is conducted on Llava-1.5-7B with five ImageNet classes. To support the generality of this geometric mechanism, this section extends the latent feature analysis to multiple MLLMs (Llava, Qwen, and Intern) and varying numbers of classes (from 2 to 9). We calculate the mean cosine similarity of PCA-reduced penultimate-layer features across four regimes: **Clean vs. Perturbed** (cross-set similarity between clean and perturbed features), **Intra-Clean** (pairwise similarity among clean features), **Intra-Perturbed** (pairwise similarity among perturbed features), and **Intra-Subcluster** (pairwise similarity within a subcluster of perturbed features sharing the same target semantics).

As shown in Figure 13, the high intra-group similarity and strongly negative inter-group similarity confirm that clean and perturbed feature sets occupy distinctly separated regions across all three models. Meanwhile, the intra-subcluster similarity ($\geq 0.95$) remains the highest across all models, demonstrating that perturbed images are deflected based on their semantics to form tight subclusters. These results are consistent with the findings in Section 5, validating the generality of our geometric conclusions across model structures.

## H. Additional Qualitative Examples Across MLLMs and Constraints

To further address the request for more qualitative examples spanning various constraints and MLLMs, we present an extended gallery in Figure 14. Similar to Figure 7, the gallery demonstrates that a single SAUP can simultaneously bind two distinct targets ("promontory" and "crayfish") to different visual concepts, switching the output between them depending on the input image. Importantly, this extended visualization spans three representative MLLMs (LLaVA, Qwen, and InternVL) under two different perturbation constraints, illustrating the generality of our attack across architectures and perturbation budgets.

Across all six model–constraint combinations, the perturbed inputs are consistently routed to the intended target captions, while the clean counterparts elicit semantically faithful descriptions. This confirms that the semantic-aware binding mechanism is not specific to a particular model or constraint setting but generalizes across diverse MLLM backbones and perturbation regimes, complementing the quantitative results presented in the main paper.

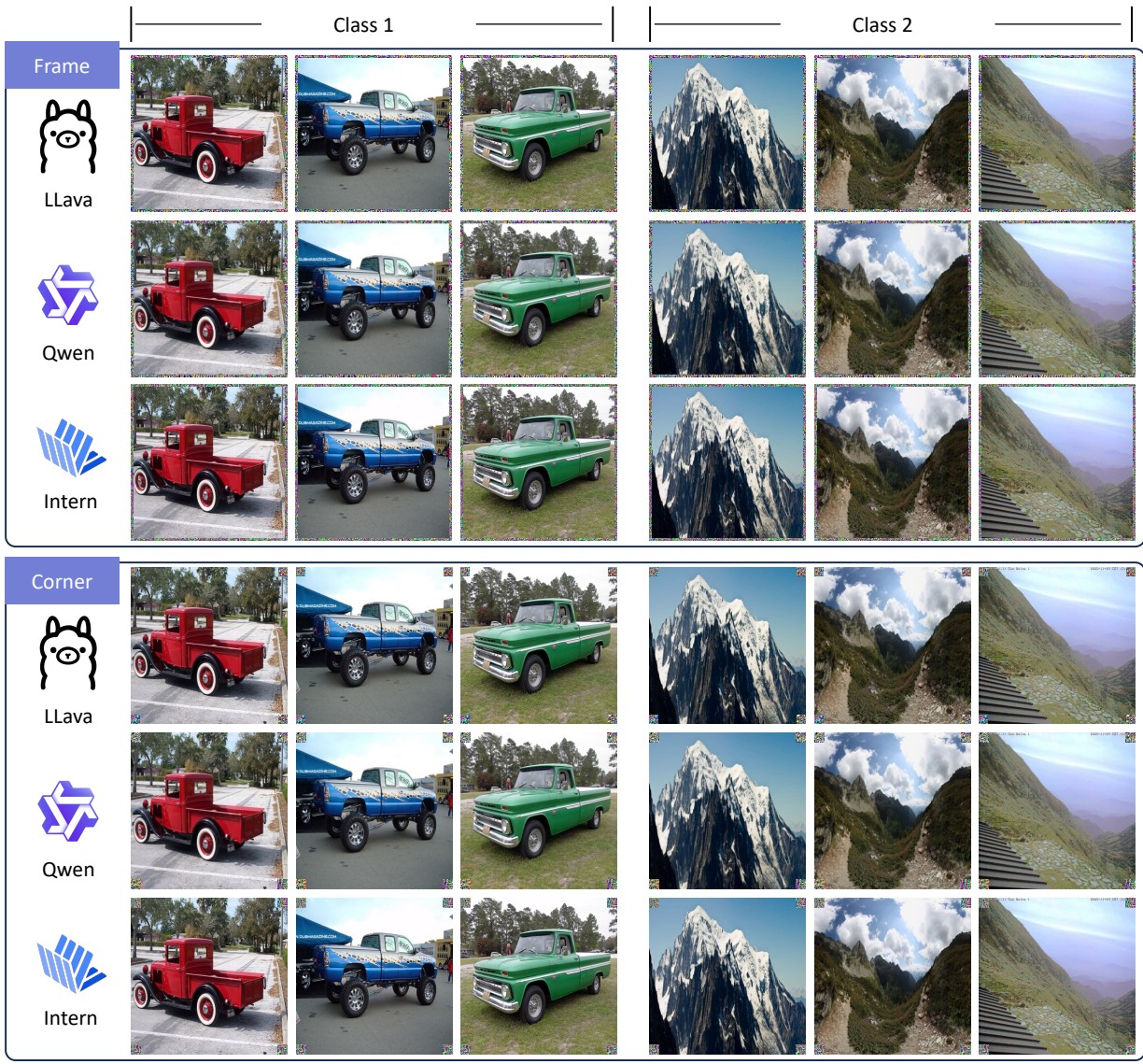

*Figure 14.* Extended qualitative gallery of SAUP across three MLLMs (LLaVA, Qwen, and InternVL) under two perturbation constraints. A single SAUP binds two targets, "promontory" and "crayfish", to distinct visual concepts. Across all model–constraint combinations, the perturbed images are consistently routed to the intended target captions, confirming that the proposed semantic-aware binding mechanism transfers across MLLM backbones and perturbation regimes.

