# OpenReview forum: "Semantic Router: On the Feasibility of Hijacking MLLMs via a Single Adversarial Perturbation"
_ICML.cc/2026/Conference — ICML 2026 regular_

### Official Review · Reviewer_yGFf · 2026-03-03

**Soundness:** 3
**Presentation:** 4
**Significance:** 3
**Originality:** 4
**Overall Recommendation:** 5
**Confidence:** 4

**Summary:**

This paper investigates a novel adversarial threat against Multimodal Large Language Models (MLLMs) termed "Semantic-Aware Hijacking." Unlike traditional Universal Adversarial Perturbations that map various inputs to a single target (many-to-one), the proposed perturbation acts as a "semantic router." It exploits a single perturbation to force the model to generate distinct, attacker-defined targets conditioned specifically on the intrinsic semantics of the input image (many-to-many).

**Compliance With Llm Reviewing Policy:**

Affirmed.

**Ethical Review Concerns:**

No ethical concerns.

**Final Justification:**

The author's response addressed my concerns, so I will maintain my positive score.

**Key Questions For Authors:**

* Could the authors include a more diverse set of case studies, particularly across different perturbation constraints and model architectures?

* Could the authors explain why the error bars are relatively large in Figure 6?

**Limitations:**

yes

**Strengths And Weaknesses:**

Strengths:

* The proposed many-to-many perturbation is an interesting and under-explored topic. By enabling conditionally triggered adversarial perturbations, this work represents a meaningful step beyond traditional UAPs.

* The theoretical framework, particularly the geometric explanations of Dominant Shift and Semantic Deflection, is intuitive and well-articulated. It is also encouraging to see the paper provide an empirical validation of this theory.

* Extensive evaluations demonstrate high attack success rates, providing strong empirical evidence for the method's effectiveness.

Weaknesses:

* The evaluation would benefit from more qualitative examples (similar to Figure 7) across various perturbation constraints and MLLMs.

* Figure 6 exhibits notably large error bars. The authors should explicitly discuss the underlying causes within the main text.

* There seems to be a typo in Section 3. The text uses $e^{(i)}$ and $e^{(j)}$, but given the context, the authors may refer to $\hat{z}^{(i)}$ and $\hat{z}^{(j)}$?

* The visualization methodology in Figure 4 is unclear. The authors should specify the dimensionality reduction technique (e.g., t-SNE, PCA).

---

> ### Author Rebuttal · Authors · 2026-03-31
>
> *Q1. The evaluation would benefit from more qualitative examples across various constraints and MLLMs.*
>
> We thank the reviewer for this valuable suggestion. In the revised manuscript, we will include an expanded qualitative gallery in the Appendix, featuring visual examples from LLaVA, Qwen, and InternVL under both Frame and Corner perturbation constraints.
>
> *Q2. The underlying causes of the large error bars in Figure 6 should be explicitly discussed in the main text.*
>
> The error bars in Figure 6 reflect performance variance across different trajectories in the RIST dataset. The relatively large magnitude of these error bars stems from two factors. First, the RIST is intentionally curated to be highly diverse, meaning the different trajectories encompass a wide range of real-world environmental conditions, such as varying lighting and backgrounds. This inherent data variance naturally induces fluctuations in attack success rates across trajectories. Second, as a fine-grained dataset, the exact semantic granularity is not completely uniform across all trajectories, leading to variation in the generalization difficulty of the perturbation. We will incorporate this explanation in Section 6.2 of the revised manuscript.
>
> *Q3. Typo regarding proxy target embeddings*
>
> We sincerely thank the reviewer for the meticulous reading and apologize for the confusion. We will revise the text to consistently use $\hat{z}^{(i)}$ and $\hat{z}^{(j)}$ for the Proxy Target Embeddings, and carefully audit the surrounding mathematical notation to ensure overall consistency and clarity.
>
> *Q4. The dimensionality reduction technique used in Figure 4 should be specified.*
>
> We thank the reviewer for pointing this out. The visualization in Figure 4 was generated using Principal Component Analysis (PCA) applied to penultimate-layer embeddings. We specifically chose PCA over nonlinear methods such as t-SNE or UMAP because it preserves the global geometric structure of the latent space, including relative distances and inter-cluster relationships. Nonlinear methods, while useful for visualization, can distort these structural properties and would therefore misrepresent the geometric relationship between Dominant Shift and Semantic Deflection. We will clarify this detail in the revised manuscript.

---

> > ### Author Rebuttal · Reviewer_yGFf · 2026-04-03
> >
> > Thanks for your response.

---

### Official Review · Reviewer_WJft · 2026-03-11

**Soundness:** 2
**Presentation:** 3
**Significance:** 2
**Originality:** 3
**Overall Recommendation:** 3
**Confidence:** 4

**Summary:**

This paper studies whether a single universal perturbation can hijack multiple stateless MLLM decisions and route different visual semantics to different attacker-defined targets. It proposes SAUP and the SORT optimization, and explains the attack with a latent-space view based on dominant shift and semantic deflection. Experiments on three MLLMs and two datasets, including the new RIST benchmark, show that this attack is feasible in the digital white-box setting.

**Compliance With Llm Reviewing Policy:**

Affirmed.

**Final Justification:**

I still have some reservations about the paper. However, after considering the other reviews and responses, I agree the paper has merit and would not be opposed to accepting it.

**Key Questions For Authors:**

How's transferability across different model architectures?

**Limitations:**

Yes, at the end of the supplementary material.

**Strengths And Weaknesses:**

Pros:
1. The paper defines a new many-to-many UAP setting for MLLMs, where one perturbation routes different input semantics to different attacker-chosen targets.
2. The paper gives a concrete latent-space explanation and also validates dominant shift, semantic deflection, and target alignment with feature visualization.
3. The experiments cover three MLLMs, two datasets, long-output control, different numbers of targets, prompt variation, and ablations, with strong results in the digital white-box setting.


Cons
1. The main analysis is based on a first-order Taylor expansion around the perturbation, and the paper explicitly says the bound ignores higher-order terms. For a highly nonlinear MLLM with autoregressive generation, this seems too weak to fully support the claimed practical feasibility.
2. The geometric story is validated only on LLaVA-1.5-7B, with five ImageNet classes, using penultimate-layer features. This is not enough to support that the same mechanism generally explains the attack across different MLLMs and settings.
3. The paper studies the attack in a digital white-box setting, and even its own ablation shows that removing NSO causes a catastrophic drop. This suggests the method may work mainly because of strong optimization access and careful reparameterization, rather than because the attack is broadly feasible.

---

> ### Author Rebuttal · Authors · 2026-03-31
>
> *Q1. For highly nonlinear MLLMs, a first-order Taylor expansion bound seems too weak to support the claimed practical feasibility.*
>
> We sincerely thank the reviewer for pointing out the gap between MLLM nonlinearity and first-order expansion, which provides a valuable opportunity to solidify our theoretical foundation.
>
> We agree that MLLMs are globally highly nonlinear, but as the theoretical explanation for adversarial examples (Goodfellow et al., 2014), these models still exhibit **local linearity**. The principle underlying local linearity is that, despite complex architectures, neural network behavior can be well approximated as linear within a small neighborhood of the input space. Since we have demonstrated that SAUP dominates the input embedding space (Figure 4(a)), inputs with the same perturbation reside in a local neighborhood where this linearity approximation remains valid. This provides a principled justification for applying the first-order Taylor expansion.
>
> Notably, the role of the first-order expansion is not to provide a zero-error prediction of model outputs, but to establish a theoretical framework that explains how a single perturbation can map semantically varying inputs to distinct targets. Through this formalization of "Semantic Deflection," we aim to provide theoretical insight and advance the mechanistic understanding of this phenomenon.
>
> Our experimental results further validate the practical feasibility of SAUP. As shown in Figure 5 and Figure 7, when the same perturbation is applied to diverse images, the MLLM acts as a "semantic router," generating distinct, predefined targets based entirely on input semantics.
>
> *Q2. The geometric story is validated only on LLaVA-1.5-7B using five classes. This is not enough to support a general mechanism across different MLLMs.*
>
> We thank the reviewer for this constructive feedback, which motivates us to broaden our empirical validation.
>
> To support the generality of the geometric mechanism, we extend our latent feature analysis to multiple MLLMs and varying numbers of classes (from 2 to 9). We report the mean cosine similarity of PCA-reduced features for LLaVA, Qwen, and InternVL:
>
> | Feature Similarity           | LLaVA | Qwen  | InternVL |
> | ---------------------------- | ----- | ----- | -------- |
> | Intra-Clean                  | 0.68  | 0.56  | 0.64     |
> | Intra-Perturbed              | 0.70  | 0.69  | 0.71     |
> | Inter (Clean vs. Perturbed)  | -0.70 | -0.62 | -0.67    |
> | Intra-Subcluster (Perturbed) | 0.96  | 0.96  | 0.95     |
>
> The high intra-group similarity and strongly negative inter-group similarity confirm that clean and perturbed feature sets occupy distinctly separated regions. Meanwhile, the intra-subcluster similarity (0.95+) remains the highest across all models, demonstrating that perturbed images are deflected based on their semantics to form tight subclusters. These results are consistent with the findings in Section 5, validating the generality of our geometric conclusions.
>
> We will incorporate these results in the revised paper.
>
> *Q3. The attack relies heavily on strong optimization access and may have limited transferability across different model architectures.*
>
> We acknowledge that our current attack, in line with existing image hijacking methods, operates under a white-box assumption and exhibits limited transferability. We argue that the main contribution of this paper is demonstrating a many-to-many perturbation that acts as a "semantic router" across inputs with diverse semantics. Establishing digital white-box feasibility is the necessary and critical first step in characterizing this new class of vulnerability.
>
> While physical and black-box deployments remain challenges, SAUPs could still pose meaningful risks in real-world settings. For instance, an attacker can target an open-source MLLM and embed the perturbation into a web advertisement, which can effectively hijack agents built on this model.
>
> We will explicitly discuss this white-box limitation in the revised manuscript.
>
> *Q4. Removing NSO causes a catastrophic drop.*
>
> We apologize for any confusion regarding the ablation result. We would like to clarify that the combination of NSO and SSO is the key to achieving optimal performance, where the NSO acts as the bedrock for the overall optimization. As a core component of the proposed method, it is reasonable and expected to observe a significant performance deduction when removing NSO in the ablation study. This drop serves as evidence that NSO is not a supplementary trick, but a necessary foundation for perturbation optimization.

---

> > ### Author Rebuttal · Reviewer_WJft · 2026-04-02
> >
> > The new results helped address some of my concerns. However, I still have questions regarding the statement "We agree that MLLMs are globally highly nonlinear, but as the theoretical explanation for adversarial examples (Goodfellow et al., 2014), these models still exhibit local linearity." Earlier work suggests that relatively smaller models (by today’s standards) can often be effectively approximated by locally linear surrogates within a meaningful neighborhood (such as LIME-based methods). However, my understanding is that these approaches appear to struggle with modern models. Can you provide more theoretical/empirical analysis to support the claim? If so, it would also be important to study the constraints/limitations/errors from both theoretical and empirical perspectives, for practical applications.

---

> > > ### Author Response · Authors · 2026-04-04
> > >
> > > We sincerely thank the reviewer for this insightful comment. We fully agree that modern MLLMs are highly nonlinear, and analyzing the constraints and limitations of our linear approximation is essential.
> > >
> > > To quantify the approximation error, we formulate the image feature as $z=\phi(\delta+\alpha\cdot v)$, where $\delta$ is the adversarial perturbation, $v$ is the normalized input image vector, and $\alpha$ controls the step magnitude. We compare the true latent feature with its first-order approximation around $\delta$ using two metrics:
> > >
> > > $$RelativeError = \frac{\left\|\left\|\phi(\delta+\alpha v)-\left(\phi(\delta)+\alpha J_\delta(v)\right)\right\|\right\|_2}{\left\|\left\|\phi(\delta+\alpha v)\right\|\right\|_2}$$
> > >
> > > $$FeatureSimilarity = \cos\left(\phi(\delta+\alpha v), \phi(\delta)+\alpha J_\delta(v)\right)$$
> > >
> > > where $J_\delta(v)$ is the Jacobian-vector product at $\delta$ along direction $v$. We evaluate these metrics on three MLLMs (Llava, Qwen, and InternVL), progressively increasing the magnitude $\alpha$ until a significant drop in loss occurs. The results are shown below:
> > >
> > > | Model | Metric | $\alpha=$ 0 | 0.1 | 0.2 | 0.5 | 1 | 2 | 5 | 10 | 20 | 50 | 100 | 200 |
> > > |---|---|---:|---:|---:|---:|---:|---:|---:|---:|---:|---:|---:|---:|
> > > | Llava | Relative Error | 0.00 | 0.05 | 0.08 | 0.13 | 0.20 | 0.39 | 0.44 | 0.63 | 0.80 | 0.91 | - | - |
> > > |  | Feature Similarity | 1.00 | 1.00 | 1.00 | 0.99 | 0.98 | 0.91 | 0.90 | 0.80 | 0.66 | 0.53 | - | - |
> > > |  | Loss | 7.51 | 7.38 | 6.66 | 7.50 | 7.37 | 6.60 | 7.42 | 7.07 | 6.18 | 2.47 | - | - |
> > > | Qwen | Relative Error | 0.00 | 0.21 | 0.29 | 0.40 | 0.49 | 0.60 | 0.68 | 0.70 | 0.73 | 0.85 | - | - |
> > > |  | Feature Similarity | 1.00 | 0.98 | 0.96 | 0.92 | 0.88 | 0.82 | 0.77 | 0.75 | 0.73 | 0.64 | - | - |
> > > |  | Loss | 1.01 | 1.01 | 1.06 | 0.92 | 1.09 | 1.29 | 1.10 | 0.81 | 0.79 | 0.34 | - | - |
> > > | InternVL | Relative Error | 0.00 | 0.00 | 0.01 | 0.01 | 0.03 | 0.16 | 0.21 | 0.19 | 0.26 | 0.41 | 0.59 | 0.76 |
> > > |  | Feature Similarity | 1.00 | 1.00 | 1.00 | 1.00 | 1.00 | 0.97 | 0.97 | 0.98 | 0.97 | 0.92 | 0.82 | 0.72 |
> > > |  | Loss | 1.65 | 1.73 | 1.73 | 1.73 | 1.73 | 1.63 | 1.74 | 1.74 | 1.77 | 1.83 | 1.88 | 0.92 |
> > >
> > >
> > > These results clarify the boundaries of our theoretical analysis and provide two key insights:
> > >
> > > 1. **Validity of Local Linearity.**
> > > Although modern MLLMs are globally highly nonlinear, all models exhibit clear local linearity for sufficiently small $\alpha$. For example, on InternVL, when $\alpha \leq 1$, the Relative Error remains negligible ($0.00\sim0.03$) and the Feature Similarity stays at $1.00$. Even at $\alpha=2$, the approximation is still accurate (error $0.16$, similarity $0.97$).
> > >
> > >     From a theoretical perspective, the non-linearity of modern MLLMs still stems from activation functions (e.g., ReLU, GELU) and other piecewise non-linear operations. Within a sufficiently small neighborhood, input changes do not cross the nonlinear boundaries of most neurons, thereby preserving local linearity.
> > >
> > > 2. **Limitations of Linear Approximation.**
> > > As the reviewer insightfully noted, as the magnitude $\alpha$ grows, the local linear assumption introduces non-negligible error. For instance, at $\alpha = 50$, the Relative Error for Qwen reaches $0.85$. However, the Feature Similarity remains moderately aligned ($0.64$), indicating that the first-order approximation transitions from serving as a precise quantitative tool to providing qualitative guidance.
> > >
> > > We deeply appreciate the reviewer for raising this critical question, which has prompted us to strengthen the theoretical foundation and clarify its limitations. We will incorporate a section titled "Empirical Analysis of Theoretical Limitations" in the revised manuscript, presenting these results and insights to enhance the overall rigor of this paper.

---

### Official Review · Reviewer_yGQG · 2026-03-13

**Soundness:** 3
**Presentation:** 3
**Significance:** 3
**Originality:** 3
**Overall Recommendation:** 5
**Confidence:** 2

**Summary:**

This paper introduces a novel security threat targeting multimodal large language models: semantic-aware hijacking. Unlike traditional universal adversarial perturbations, the proposed semantic-aware universal perturbations function as a "semantic router." Within a single perturbation framework, they can steer the model's output to multiple distinct, attacker-predefined target texts or instructions based on the semantic content of the input image. The paper validates the fundamental feasibility and effectiveness of this attack on three mainstream MLLMs through theoretical analysis, algorithmic innovation, dataset construction, and extensive experiments.

**Compliance With Llm Reviewing Policy:**

Affirmed.

**Key Questions For Authors:**

See weaknesses

**Limitations:**

Yes

**Strengths And Weaknesses:**

## Strengths:
- Innovation and Application Potential:​ The paper's core contribution is innovative, systematically exploring and implementing a "one-to-many" semantic-aware hijacking. This attack pattern demonstrates clear and promising application potential, especially in autonomous driving, robotics, and other scenarios where intelligent agents rely on MLLMs for sequential, stateless decision-making. A well-designed perturbation could, in theory, "program" the behavioral sequence of an agent, posing a severe threat.
- Well-Structured Paper:​ Particularly, the proposed SORT algorithm and the annotated RIST dataset provide valuable tools and benchmarks for future research.
- Warning Significance:​ This work serves as a timely reminder for researchers and developers in related fields to be vigilant about such advanced security issues. It shows that even if individual decisions appear stateless, their concatenation can be systematically manipulated by a unified perturbation, raising new and higher demands for MLLM security assessment and robustness design.
## Weaknesses:
- Practicality Concerns Regarding Fine-Grained Attacks:​ Although the paper presents attack results for fine-grained semantics on the RIST dataset, its definition of fine-grained (e.g., "highway" vs. "sidewalk") might still be considered coarse in the complex, continuous decision-making scenarios of the real world. A deeper exploration of the attack possibility and limits under truly extreme fine-grained conditions, rather than relatively broad category distinctions, would be more desirable for researchers.
- Idealized Experimental Settings:​ All experiments are conducted in a digital white-box environment with constrained perturbations (borders, corners), which significantly differs from real-world deployment.

---

> ### Author Rebuttal · Authors · 2026-03-31
>
> *Q1. The definition of "fine-grained" in RIST might be considered coarse for real-world continuous decisions. A deeper exploration of extreme fine-grained conditions is desired.*
>
> We appreciate the reviewer's insightful point regarding the necessity of analyzing extreme fine-grained conditions. The ability to handle such scenarios is a critical measure of the attacker's flexibility. This is exactly our core motivation for curating the RIST dataset.
>
> Since "extreme fine-grained" is not formally defined in the review, we interpret it as the ability to distinguish between highly similar, instance-level visual inputs. Under this interpretation, we clarify that RIST has already considered this level of granularity, distinguishing not only between broad categories (e.g., "highway" vs. "sidewalk"), but also between visually similar instances (e.g., "this specific sidewalk" vs. "another specific sidewalk").
>
> To quantitatively characterize this semantic granularity, we compute the CLIP Score for RIST images to measure the similarity of intra-class and inter-class image pairs.
>
> | Semantic Similarity | Metric | Intra-Class | Inter-Class |
> |---------------------|--------|-------------|-------------|
> | AutoDriving         | min    | 0.78        | 0.42        |
> |                     | max    | 0.94        | 0.88        |
> |                     | mean   | 0.90        | 0.68        |
> | RoboTasking         | min    | 0.75        | 0.66        |
> |                     | max    | 0.92        | 0.80        |
> |                     | mean   | 0.84        | 0.73        |
>
> These results confirm the fine-grained nature of RIST. Notably, the mean inter-class similarity reaches approximately 0.70, with maximum values exceeding 0.80. This high inter-class similarity requires SAUP to exploit highly subtle visual differences to route inputs to distinct target labels. We will include these quantitative results and clarifications in the revised paper.
>
> We believe this instance-level granularity covers the vast majority of attack scenarios. Should the reviewer wish to explore more extreme conditions,  we would be pleased to provide additional analyses in the revision.
>
> *Q2. The digital white-box setting with constrained perturbations differs significantly from physical real-world deployment.*
>
> We acknowledge that our work relies on the white-box assumption, and physical deployment remains a challenge for future work. Nevertheless, the main contribution of this paper is to demonstrate that semantic-aware universal perturbations are feasible, which still carries potential security implications for real-world applications.
>
> For instance, with the rise of MLLM-based agents for web and smartphone interaction, an attacker can target a widely deployed open-source MLLM and embed a crafted SAUP into a web advertisement. In such a scenario, the SAUP could steer the agent to first upload private data and then execute a malicious "submit" action. We hope SAUP serves as a starting point to motivate future research on semantic hijacking in more complex agentic environments.
>
> We will explicitly discuss this limitation in the revised paper.

---

> > ### Author Rebuttal · Reviewer_yGQG · 2026-04-03
> >
> > My expertise in visual LLMs is limited, so I am uncertain whether the issues noted by fellow reviewers seem to be fully resolved.

---

### Decision · Program_Chairs · 2026-04-30

**Decision:**

Accept (regular)

**Comment:**

The paper introduces semantic-aware universal perturbations (SAUP) that act as a "semantic router", which enables a single perturbation to map different input semantics to distinct attacker-defined outputs.
It is recognized the novelty and significance of this many-to-many attack formulation. Author(s) have provided solid empirical results across multiple MLLMs and a useful benchmark dataset.

Although there are some concerns on the geometric explanation and real-world applicability, author(s) have substantially clarifies these issues by adding empirical analysis of approximation limits and broader validation in rebuttal.

Overall, this paper has strong novelty, practical security implications and adequate technical experiments.